# Convolutional Neural Network-Based Automatic Analysis of Chest Radiographs for the Detection of COVID-19 Pneumonia: A Prioritizing Tool in the Emergency Department, Phase I Study and Preliminary “Real Life” Results

**DOI:** 10.3390/diagnostics12030570

**Published:** 2022-02-23

**Authors:** Davide Tricarico, Marco Calandri, Matteo Barba, Clara Piatti, Carlotta Geninatti, Domenico Basile, Marco Gatti, Massimiliano Melis, Andrea Veltri

**Affiliations:** 1AITEM Artificial Intelligence Technologies Multipurpose s.r.l., Corso Castelfidardo 36, 10129 Turin, Italy; davide.tricarico@unito.it (D.T.); massimiliano.melis@aitemsolutions.com (M.M.); 2Department of Mathematics “G. Peano”, University of Turin, Via Carlo Alberto 10, 10123 Turin, Italy; 3Diagnostic and Interventional Radiology Unit, Oncology Department, San Luigi Gonzaga University Hospital, University of Turin, Regione Gonzole 10, 10043 Orbassano, Turin, Italy; marco.calandri@unito.it (M.C.); clara.piatti@hotmail.it (C.P.); carlotta.geninatti@unito.it (C.G.); domenico.basile@unito.it (D.B.); andrea.veltri@unito.it (A.V.); 4Radiology Unit, Department of Surgical Sciences, University of Turin, Città della Salute e della Scienza di Torino, Corso Bramante, 88/90, 10126 Turin, Italy; m.gatti@unito.it

**Keywords:** artificial intelligence, Convolutional Neural Network (CNN), deep learning, coronavirus disease 2019 (COVID-19), chest X-ray, prioritization

## Abstract

The aim of our study is the development of an automatic tool for the prioritization of COVID-19 diagnostic workflow in the emergency department by analyzing chest X-rays (CXRs). The Convolutional Neural Network (CNN)-based method we propose has been tested retrospectively on a single-center set of 542 CXRs evaluated by experienced radiologists. The SARS-CoV-2 positive dataset (*n* = 234) consists of CXRs collected between March and April 2020, with the COVID-19 infection being confirmed by an RT-PCR test within 24 h. The SARS-CoV-2 negative dataset (*n* = 308) includes CXRs from 2019, therefore prior to the pandemic. For each image, the CNN computes COVID-19 risk indicators, identifying COVID-19 cases and prioritizing the urgent ones. After installing the software into the hospital RIS, a preliminary comparison between local daily COVID-19 cases and predicted risk indicators for 2918 CXRs in the same period was performed. Significant improvements were obtained for both prioritization and identification using the proposed method. Mean Average Precision (MAP) increased (*p* < 1.21 × 10^−21^ from 43.79% with random sorting to 71.75% with our method. CNN sensitivity was 78.23%, higher than radiologists’ 61.1%; specificity was 64.20%. In the real-life setting, this method had a correlation of 0.873. The proposed CNN-based system effectively prioritizes CXRs according to COVID-19 risk in an experimental setting; preliminary real-life results revealed high concordance with local pandemic incidence.

## 1. Introduction

In 2020, the whole world was turned upside down by the COVID-19 pandemic. Italy was one of the first Western countries to be heavily affected and up to now, June 2021, more than 4,000,000 cases have been recorded, with more than 125,000 deaths [1].

A key factor for the healthcare system response to the COVID-19 pandemic is the application of a correct diagnostic work up and triage of COVID-19 suspected cases. The RT-PCR test is the gold standard diagnostic tool for the identification of SARS-CoV-2. It has a high specificity, but a relatively low sensitivity (54–73%) [2]; furthermore, results can take hours to come out, creating logistic issues in the emergency departments [3].

In this setting, radiology plays a crucial role [4]. In particular, chest X-rays (CXRs) are used during triage due to the large availability of portable units, that are easy to use and fast to clean. This approach has been endorsed by the main radiological societies worldwide since the beginning of the pandemic [5,6]. However, CXR limits are well-known. Indeed, preliminary data suggest that CXRs may not be sufficiently sensitive for the detection of COVID-19 lung disease, reducing the potential clinical and diagnostic impact of this exam [7].

In this emergency department scenario, the availability of a tool that can prioritize the most suspicious CXRs may be of great value in order to promptly identify COVID-19 cases and to reduce promiscuity in the waiting room.

With the rise of machine and deep learning technologies [8,9], more and more methods have been proposed to tackle tasks in the analysis of CXRs [10]. The availability of large public datasets of CXRs [11,12,13] and the COVID-19 pandemic gave a great stimulus to research on CNN for early detection of SARS-CoV-2 positive cases, with many new approaches proposed over the last year. A large majority of the proposed solutions tackled disease identification, based on deep learning algorithms [11,14,15,16,17,18], providing some levels of interpretability [19,20,21]. Other studies approached different tasks like quantification of infection severity [22,23,24], segmentation of image [22,25], prediction of disease evolution [26] and image synthesis [27]. The aim of our study was to design a system for the prioritization, based on COVID-19 infection likelihood, of CXRs to support the diagnostic workflow [28,29].

However methodological flaws and underlying bias are common among these papers, as recently highlighted [30]. Being aware of the limitations of the first AI-based experiences on this topic, the aim of this study was to evaluate, both in an experimental model as well as in real-life, the feasibility and the efficacy of a CNN-based prioritizing system, to analyze the capability of identifying suspected CXRs and to put them upfront in the radiologists’ worklist.

## 2. Materials and Methods

The study was approved by the IRB in full accordance with the Helsinki declaration and its later amendments.

The effectiveness of our method was tested in two different settings. Firstly, the algorithm was applied to a set of retrospectively analyzed CXRs to assess some performance metrics, comparing system predictions with actual COVID-19 positivity. Secondly, the system has been integrated in the hospital PACS/RIS to automatically receive and analyze all the CXRs. The predictions calculated by the system have been compared with COVID-19 local incidence.

### 2.1. Deep Learning Method

The proposed method, named AIppo, is an AI system designed to exploit the similarities between images and analyze pictures indicating how much they are similar to CXRs of COVID-19 severe cases or of healthy cases. The system is based on a deep neural network derived by DenseNet-121 architecture [31], using metric learning to define the similarities between images. Weights for the deep neural network were transferred from the ones obtained by training the same architecture on the ImageNet classification task. Fine-tuning had not been performed, so that the CNN weights used to compute performance were the ones from ImageNet. An overview of AIppo is illustrated in Figure 1.

In the pre-processing stage, after a proper resize, the U-Net lung segmentation model crops down CXRs producing a final image that just includes the lungs [32], excluding parts of the image that can add details unrelated to COVID-19, and is finally normalized by subtracting its average and dividing by its standard deviation. Figure 2 illustrates these steps more in detail: the original image (a) is resized limiting the maximum dimension to 1000 pixels, but maintaining its original aspect ratio (b). The resized picture (c) is then processed by the U-Net model (d) to extract a binary mask (e) indicating where the lungs are located. To define the final cut (g), the binary mask (e) is framed by the smallest rectangle that contains the lungs (f). The resized picture (c) is normalized by mean and variance (h,i) and, finally cropped (j) using the computed rectangular cut obtaining the final picture (k). The final image is then elaborated by a deep neural network to extract the features that represent the semantic of the picture.

The proposed system includes a database, indicated by (f) in Figure 1, that includes a set of cases from the past about the presence of COVID-19 infection which is known a priori. This set is referred to as the “reference set” and only the feature vectors extracted from the images in this set and related ground truth are memorized. When a new picture is received by the system and its feature vector is computed by the deep neural network model, the method calculates the angular similarity between the computed vector and the ones stored in the database. Then, it retrieves the 10 most similar cases and related ground truth. To compute the final score, the system counts how many of the retrieved nearest neighbors belong to the COVID-19 positive set, summing up their contribution by a weighting function that depends on the similarity value and position in the neighbors’ similarity ranking. This function decays logarithmically with the position and increases linearly with the similarity, so that the most similar cases account for most of the final score. A term to scale the score in the range 0 to 1 is also included in the function. In this way, the resulting COVID-19 risk indicator obtains higher values the more the new picture is similar to past COVID-19 positive known cases.

As the number of past known cases grows, this methodology is expected to increase its level of performance taking advantage of the natural accumulation of samples. The calculated similarity can also be used by the tool to support the physicians during the diagnostic process, taking out the most similar cases from the past for comparison.

### 2.2. Phase I Study

A set of CXRs has been provided by Università di Torino for this analysis. In this phase, we define the metrics to be estimated and the methodology to be adopted to measure the performance of the proposed solution.

#### 2.2.1. Patients’ Population and Images

A total of 542 CXR images were retrospectively collected at the Department of Radiology of the University of Torino, of which 234 were COVID-19 positive cases collected during the virus outbreak.

CXRs were considered COVID-19 positive and included in the COVID-19 group in the case of positive nasopharyngeal swabs within 24 h. The control group included 308 CXRs performed before the pandemic. Furthermore, negative cases were retrospectively reviewed by three radiologists (20, 10 and 3 years of experience) to discriminate patients without any lung disease from those with positive radiological findings (oncological disease, pneumonia of different origins…). Only postero-anterior views were included in the analysis.

The final set of images includes 542 samples composed as the following:146 completely healthy patients234 COVID-19 positive cases162 subjects affected by other diseases

Figure 3 illustrates data selection and labelling adopted for phase 1.

#### 2.2.2. Performance Metrics and Statistical Analysis

The proposed methodology was designed to analyze each CXR and provide a COVID-19 risk indicator to prioritize diagnostic workflow. The presence of a high value for this indicator can also be used to identify cases in need of urgent care. To measure the effectiveness of our proposed method, we calculated the performance metrics illustrated in Table 1. All the metric values range from 0% to 100%. Given a set of cases, we can sort it by the predicted scores and compute the Mean Average Precision (MAP) metric. The MAP is an evaluation measure for the information retrieval system used to assess how well the system is performing in providing sorted lists of results to a given set of queries. This metric assumes values in the range between 0 and 1; it considers the order in which the returned results are presented. It assumes value equals to 1 when all the relevant results are placed on top. In our case, the order of returned results was obtained sorting the cases by the predicted COVID-19 risk indicator and relevant cases were the ones belonging to the COVID-19 positive class. Given the score of each analyzed case, we could identify the more critical ones if they were higher than a threshold and estimated sensitivity and specificity as measures of classification performance, by comparison with actual COVID-19 positivity. Sensitivity and specificity are evaluation metrics for classification tasks to assess how well the model is identifying the class membership of a given query image; these metrics indicate, respectively, the portion of actual COVID-19 positive and negative cases correctly identified. They assume values in the range between 0 and 1, where 1 indicates all the cases were assigned to the correct class.

To estimate each metric, we iterated 5-fold stratified cross validation by 6 times. At each iteration, samples were randomly shuffled, and 5 folds were extracted. Four folds out of five were used as the reference set, while the remaining one was used for testing. This procedure resulted in 30 different evaluations for each metric from which the average and 95% confidence interval were computed based on the Gaussian distribution hypothesis (Shapiro test *p*-value > 0.05). To further investigate the prediction power of the proposed method, we divided data by age group and gender to highlight significant differences.

#### 2.2.3. COVID-19 Risk Indicator Statistical Analysis

In literature, different approaches have been proposed to provide a semi-quantitative assessment of infection severity when analyzing CXRs. In 2015, a simple 5-level-rating was proposed in [33] to facilitate the management of clinical workflows for patients with severe acute respiratory infection. A more specific semi-quantitative COVID-19 severity rating has been proposed in [34]; this score is based on the presence on CXR of radiological findings to support the identification of biomarkers discriminatory for COVID-19. The resulting index is defined in a range from 0 to 18, where 18 represents the highest level of severity. In a similar way, our method predicted a COVID-19 risk indicator, but it was designed to represent the likelihood of infection rather than the level of infection.

In [35,36], the relationship between the severity score from [34] and patient age and gender has been analyzed, reporting the same results for age and different ones for gender. In [35], significantly higher scores were determined for male patients in the range between 50 and 80 years old, but this finding has not been confirmed in [36]. In both papers, a low positive correlation with age was found.

To compare our results with the ones from previously introduced works, we computed the Spearman correlation coefficient between patient age and COVID-19 risk indicator, the latter being computed by the proposed algorithm for SARS-CoV-2 positive patients. We made this calculation on both the whole dataset and divided by gender group.

It is important to underline that the cases’ selection criteria in the referenced studies include, apart from the RT-PCR test result, also the presence in the CXR picture of radiological findings to support the identification of biomarkers discriminatory for COVID-19. In our case, this selection was not performed. Finally, the datasets differ for sample size: in the referenced works, 734 and 1000 positive patients were provided, while 236 positive cases were available for our study.

#### 2.2.4. Reference Values

To understand if the proposed method can bring the targeted benefits, we defined reference values per each of the selected metrics. Currently, apart for particular cases identified during triage and access to the hospital, diagnostic workflow is mostly managed by first-in-first-out (FIFO) policy so that CXRs are examined by a radiologist in chronological order. To obtain reference values, we modelled this approach by a random sorting policy, resulting in MAP = 47.79% (40–48%). Radiologist diagnostic sensitivity = 61% (55–67%) has been measured according to literature data [37]. Radiologist specificity = 63% (40–89%) has been determined according to literature data [38,39,40].

### 2.3. Real-Life Preliminary Evaluation

The proposed system has been installed at the local hospital to automatically analyze CXRs. In this phase, we define how we compare prediction values with public data about local incidence of COVID-19. Figure 4 depicts the data selection and elaboration process used in phase 2.

#### 2.3.1. System Integration and Installation

In April 2021, at AOU San Luigi Gonzaga a machine running a prototype version of AIppo was installed in the hospital network. Figure 5 illustrates the system at a high level.

Three portable X-ray units, from two different vendors, adopted in the hospital during the COVID-19 pandemic have been configured to send all the produced samples to both hospital PACS and the AIppo machine. A local PACS service, running on the latter, manages the inbound stream of samples according to the communication protocol and stores the received X-rays. When new samples are received, the system excludes the ones that do not satisfy the criteria for the body part (CHEST) and position (AP/PA); images are then elaborated by the system to produce COVID-19 indicators, as described in the previous section. When the prediction is ready, this is automatically sent to the RIS, where a likelihood value appears in the radiology diagnostic queue dashboard with an associated color to provide an immediate and easy to read feedback to the radiologist (red for high-risk patients, yellow and green for medium and low risk). This visualization has been defined together with the radiologists to minimize the cognitive overhead introduced by the introduction of the AIppo prediction score in the workflow.

#### 2.3.2. Reference Data

We compared the COVID-19 scores predicted by the system with local COVID-19 incidence to check if the tool follows the trend. To do so, we analyzed the machine internal database in the period between 9 April 2021 and 4 June 2021 with public available and daily updated data of the same period from Ministero della Salute on COVID-19 incidence in Provincia di Torino [41], the local district where the AOU San Luigi Gonzaga of Orbassano is situated. Figure 6 illustrates the new cases per day in the period of interest.

#### 2.3.3. Performance Metrics and Methodology

To measure the ability of the tool to follow the pandemic trend, we calculated the correlation by means of the Pearson correlation coefficient between the 7 day rolling averages of the COVID-19 AIppo score and local pandemic incidence. Significance of correlation was evaluated according to the Two-Tailed or Nondirectional Test with alpha = 0.05.

## 3. Results

### 3.1. Baseline Characteristics

The age distribution of subjects in the positive and negative groups is illustrated in Figure 7. None of the two exhibited normality for age distribution (Shapiro–Wilk test *p*-values: 0.0012 and <1 × 10^−7^, alpha: 0.05). The two cohorts did not differ significantly in age average (positive: 67.2, negative: 66.7, Kruskal–Wallis test *p*-value: 0.792, alpha: 0.05), variance (positive: 253.7, negative: 281.6, Bartlett test *p*-value: 0.153, alpha: 0.05) and distribution (Mann–Whitney test *p*-value: 0.389, alpha: 0.05). They did not differ significantly also in gender distribution (Fisher exact test *p*-value: 0.539, alpha: 0.05). To analyze the variability of class membership (COVID-19, Other, No Finding) and age we divided the samples into homogeneous groups. When grouping by gender, the differences between distributions were not significant for class membership (Chi-square test *p*-value: 0.986, alpha: 0.05), and age, (Mann–Whitney U test *p*-value: 0.178, alpha: 0.05). Finally, we divided samples by age groups, evaluating the differences in COVID-19 positive case prevalence between males and females; this difference was significant for every age group apart from patients in the 80–89 years old range. These results are summarized in Table 2 and Table 3.

### 3.2. Phase I Study

Given the provided set of images, metrics have been estimated and compared with reference values. The estimated values, with confidence intervals, are summarized in Table 4. Each one of the metrics showed a significant increase compared with the reference values.

To investigate how performance can be influenced by age and gender, we compared distributions of MAP for different age groups, as illustrated in Table 5, by means of the Kruskal–Wallis statistical test (alpha = 0.05) and between gender groups, as illustrated in Table 6, by means of the Mann–Whitney U statistical test (alpha = 0.05). In both cases, the differences were significant, with *p*-value < 1.12 × 10^−12^ and *p*-value < 1.60 × 10^−4^, respectively. This difference suggests a possible gap in performance between the groups. Considering age groups, we can notice a higher-than-average performance for younger patients. For gender grouping, the relative gap in MAP performance, 10.72%, is partially compatible with the difference in COVID-19 positive shares, 6.25%. The gap can also be interpreted by the effect of the imbalance in the starting dataset with 305 males and 231 females.

#### COVID-19 Risk Indicator Statistical Analysis

We analyzed how the predicted COVID-19 risk indicator was influenced by the age and gender of the patient. Figure 8 and Figure 9 illustrate the distribution of predicted COVID-19 risk indicators for cases in the three labeled classes (COVID-19, Other and No Finding) for the whole and gender-grouped datasets, respectively. In the first picture, the shift in distribution for the COVID-19 positive class with respect to the others can be appreciated: No Finding is placed in the lower part of the range, while Other samples are spread in the middle. When grouping by gender, the structure is maintained with some shifts, in particular scores for No Finding and COVID-19 classes appear to be higher in the male group, with a large gap in mean value for the COVID-19 class. The abovementioned different class balance between the two genders can be a possible interpretation for this change of behavior: since the final score is computed exploiting the similarity with other cases, having a different number of available samples from one class in the reference set can impact the performance for that specific class.

We computed the Spearman correlation coefficient [Rs] between patient age and COVID-19 risk indicator computed by the proposed algorithm for patients positive for COVID-19, on both the whole dataset and divided by gender group. The results are summarized in Table 7. For the whole dataset, there was almost no correlation with age, Rs = −0.004, *p*-value = 0.884; in [35], a low positive correlation was obtained, but it was not statistically significant. Grouped by gender, the correlation result was low and positive in both cases, but not significant, Rs = 0.167 *p*-value < 5.52 × 10^−10^ for the female group, and Rs = 0.065 *p*-value < 5.13 × 10^−4^ for the male one, confirming the results in [35].

We divided the cases by age groups, and, for each age group, we studied the score distribution testing the differences between genders with the Mann–Whitney U test (alpha = 0.05). The difference was significant for four out of six age groups, highlighting a possible gender bias in model scoring. With the given class labelling we could not prove this is caused by the algorithm itself or by the underlying COVID-19 severity distribution in the data.

### 3.3. Phase 2: Real-Life Setting

In the period of interest, 2942 CXRs from the three X-rays units have been analyzed by the proposed method, providing the COVID-19 risk indication for each of the CXRs. The correlation index between the 7 day rolling averages of the AIppo score and local pandemic trend was 0.873, *p*-value < 1 × 10^−5^. Figure 10 plots the values for both pandemic trend and AIppo prediction in the considered period for Provincia di Torino.

## 4. Discussion

Our findings demonstrate the feasibility of the application of a CNN model in order to turn a FIFO workflow model into a smart prioritized workflow for the early identification of COVID-19 lung disease in the emergency department. The findings demonstrate an excellent result of the CNN-based system in the experimental setting with the MAP index moving from 43.79% to 71.75%.

A detailed analysis of results highlighted that a large part of the mispredictions comes from patients affected by diseases other than COVID-19, which can mimic SARS CoV2-positive images. Accordingly, we recomputed the MAP index excluding these cases and we obtained a MAP index of 90.37%, higher than the 61.58% measured with FIFO policy sorting.

In our preliminary analysis, we also set an alert threshold to measure the ability of the tool to identify the most critical cases. To tune this value, a specificity equal or higher than radiologists’ specificity reported in literature [38,39,40] has been targeted. By doing this, the sensitivity of 78.23% (65–92%) and specificity of 64.20% (63–66%) were obtained, in both cases higher than the defined reference values, 61.1% (55–67%) and 63% (40–89%), respectively. These results demonstrate the possibility to properly trigger alerts, enabling prompt care of critical subjects and fast application of infection containment actions.

Furthermore, in the preliminary real-life analysis based on 2942 CXRs, COVID-19 incidence curve and average day risk score calculated by the AIppo system decreased concurrently during the evaluation period.

Immediately after the COVID-19 outbreak, several experiences have been published concerning AI-based models applied to CXRs for the detection of COVID-19 lung infection with a broad range of methodologies (type of datasets, labelling, machine learning vs. deep learning approach) and aims (diagnosis vs. prognosis). As recently highlighted [30], a vast majority (72%) of AI-proposed solutions suffer from a high risk of bias coming from the dataset used in the study. The potential threats were primarily due to the following reasons:usage of public datasets with scarce control over label and image qualityarbitrary methods used to select a subset of the original dataseta large difference in characteristics of participants between COVID-19 positive and negative groupslarge imbalance between groups

A relevant portion of proposed articles put together a dataset by merging different datasets in the attempt to have a good number of positive, negative and total cases. This approach exposes to the risk of high bias the abovementioned (1) and (3). Moreover, the usage of different sources for positive and negative cases can induce the CNN models to discriminate between data sources instead of disease-related features.

To minimize the risk of introducing bias, we analyzed each of the abovementioned criteria for the provided dataset, in order to overcome limitations. As for participants’ selection, no public datasets were used and SARS-CoV-2 positive and negative CXRs have been produced in the same institution from three machines by two different vendors. Furthermore, patients included in the experimental setting tested positive for SARS-CoV-2 according to a RT-PCR test performed within 24 h from the CXR. Of note is that all CXRs were retrospectively revised by three independent radiologists and, as for data analysis and groups balance, no significant differences in statistical distributions of age and gender have been found.

To the best of our knowledge, this is the first report that includes in the analysis a preliminary real-life application. As mentioned, the proposed system can improve its performance over time because more experience is accumulated in its internal database to provide a larger base of knowledge for a more precise case comparison. This aspect is particularly favorable in the hospital setting, where a large amount of CXRs can be produced in a reasonable time making it faster to converge at higher levels of performance.

The comparative analysis between the system prediction and local COVID-19 incidence trend highlighted how the tool average prediction is consistently following the decreasing pandemic wave, providing one more element to convince us that the approach is promising.

The creation of a smart worklist that reduces the report turnaround time for critical patients using AI tools has been recently proposed in a paper by Baltruschat et al. [28]. Here, we propose a specific model for COVID-19 suspicious CXRs; the authors believe that a specific COVID-19 CXR prioritizing tool might be of great value in order to promptly identify the infected cases and to reduce promiscuity in the waiting rooms of the emergency department [42].

Our paper has some limitations. Firstly, despite being a supervised classified dataset, the number of available CXRs is limited (only 542 samples). Secondly, the data collection involved only one center. To partly mitigate the effect of the latter, the dataset is produced with different machines from different vendors, homogeneously between classes.

## 5. Conclusions

In conclusion, our study confirms the AI capability of identifying COVID-19 suspicious cases, limiting potential biases of previous series. The real-life preliminary analysis on almost 3000 CXRs suggests a promising concordance with local incidence of COVID-19 cases.

## Figures and Tables

**Figure 1 diagnostics-12-00570-f001:**
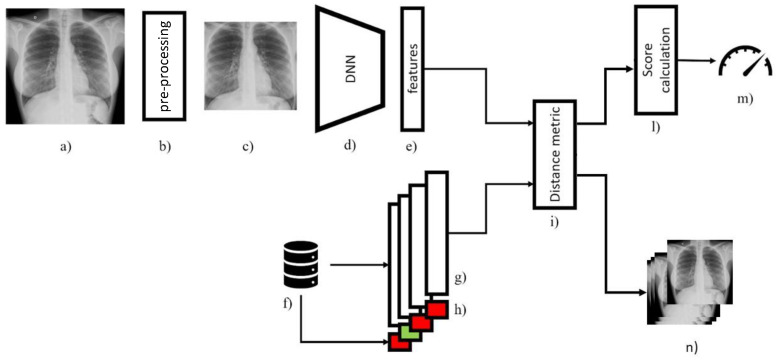
The tool architecture. The image (**a**) is pre-processed by algorithm (**b**) into the new image (**c**) which is elaborated by deep neural network (**d**) to extract features (**e**). Computer features are compared by the distance metric (**i**) with the previously extracted features (**g**) stored in the database (**f**) with their labels (**h**). The features (**g**) have been calculated on past known cases. Using the similarity information elaborated by (**i**) and labels (**h**) a COVID-19 score (**m**) is computed by stage (**l**). The set of most similar cases from the past (**n**), used for the computation of score (**m**), is returned to support doctor diagnosis and to provide interpretation of result.

**Figure 2 diagnostics-12-00570-f002:**
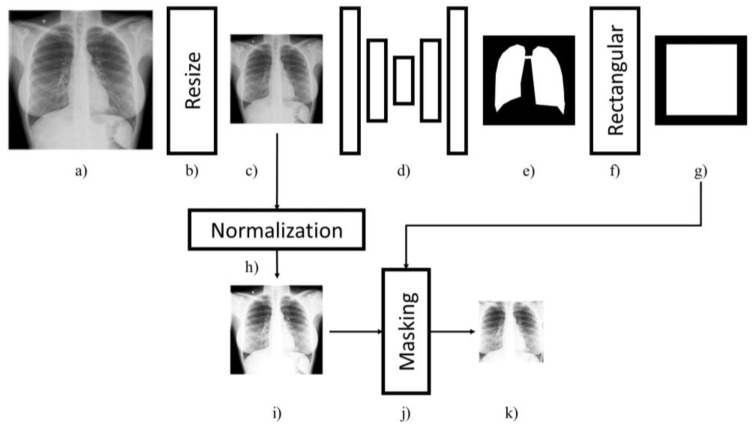
The pre-processing stage. The original image (**a**) is properly resized limiting the maximum dimension to 1000 pixels but maintaining its original aspect ratio (**b**). The resized picture (**c**) is then processed by the U-Net model (**d**) to extract a binary mask (**e**) indicating where the lungs are located. To define the final cut (**g**), the binary mask (**e**) is framed by the smallest rectangle that contains the lungs (**f**). The resized picture (**c**) is normalized by mean and variance (**h**,**i**) and finally cropped (**j**) using the computed rectangular cut to obtain the final picture (**k**).

**Figure 3 diagnostics-12-00570-f003:**
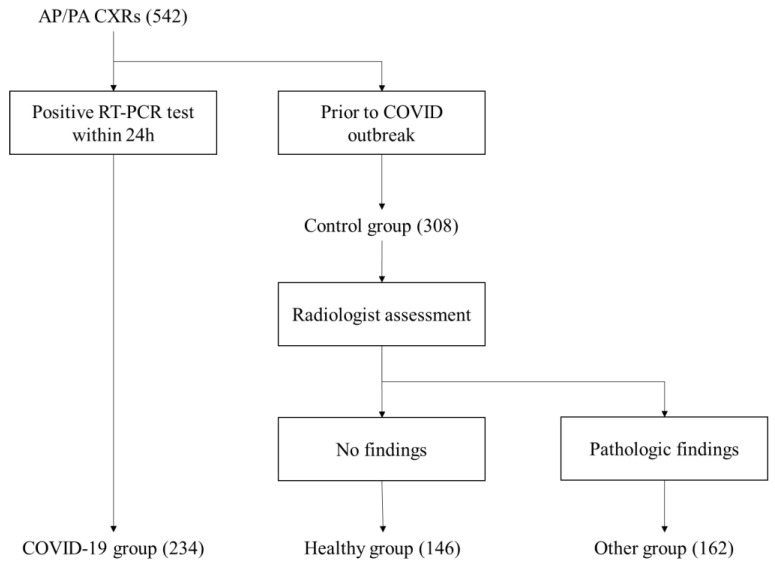
Data selection and labelling during phase 1. Images have been assigned to different groups according to RT-PCR test (for COVID-19) and assessment by a team of radiologists (for other diseases).

**Figure 4 diagnostics-12-00570-f004:**
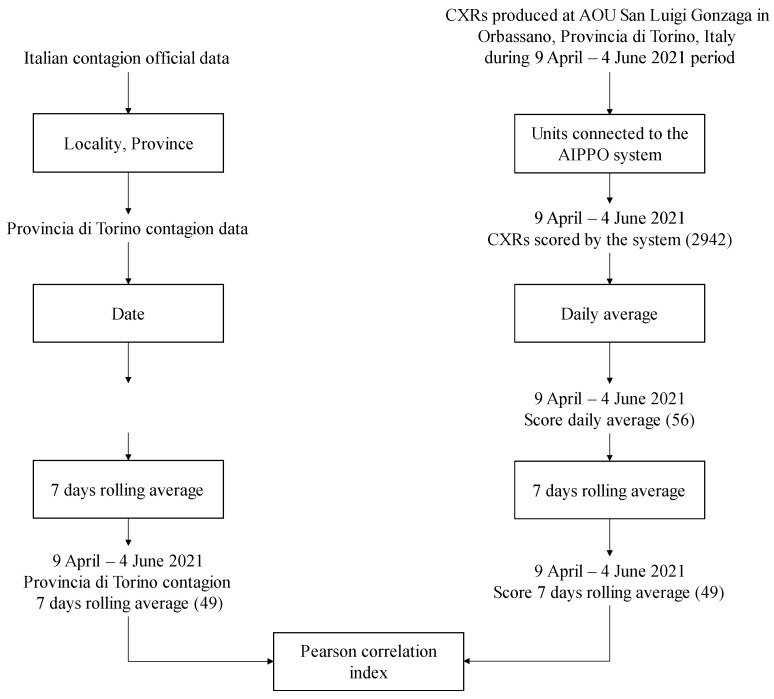
Data selection and elaboration process adopted in phase 2.

**Figure 5 diagnostics-12-00570-f005:**
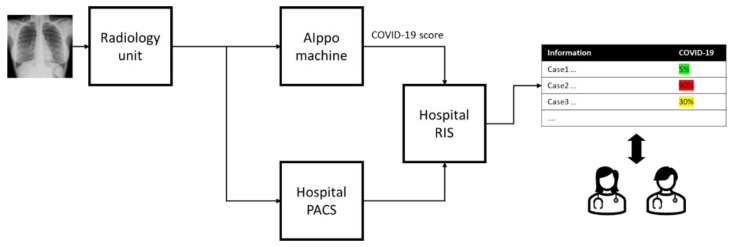
High level view on system installed at hospital for phase 2.

**Figure 6 diagnostics-12-00570-f006:**
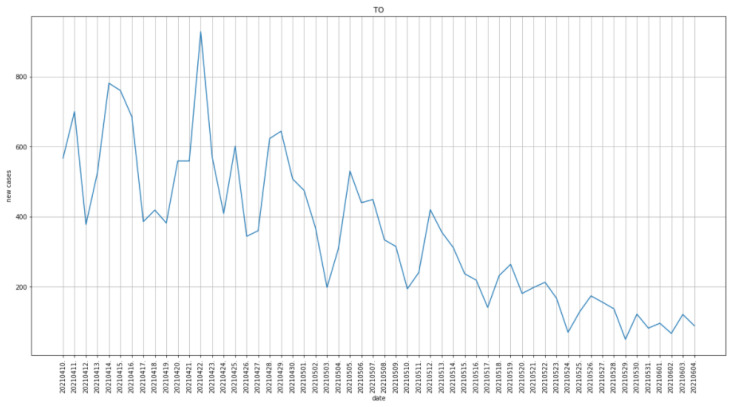
Daily new cases of COVID-19 in Provincia di Torino. Vertical axis: daily new cases, horizontal axis: date.

**Figure 7 diagnostics-12-00570-f007:**
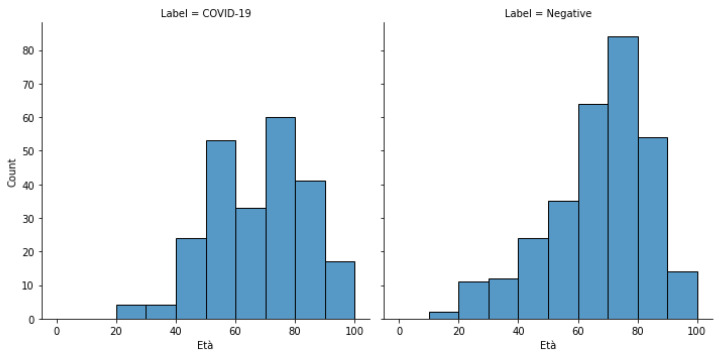
Distribution of the positive and negative groups.

**Figure 8 diagnostics-12-00570-f008:**
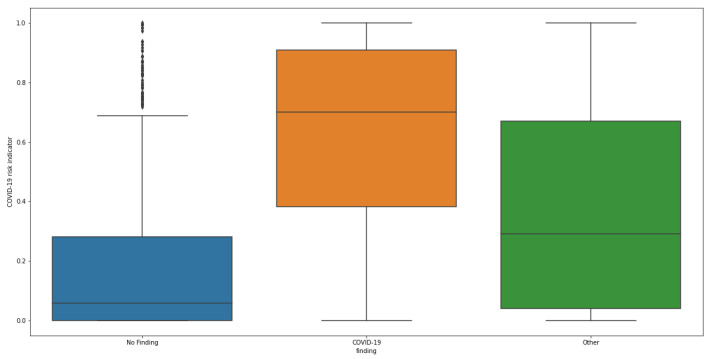
Distribution of the predicted COVID-19 risk indicator for cases belonging to the three classes.

**Figure 9 diagnostics-12-00570-f009:**
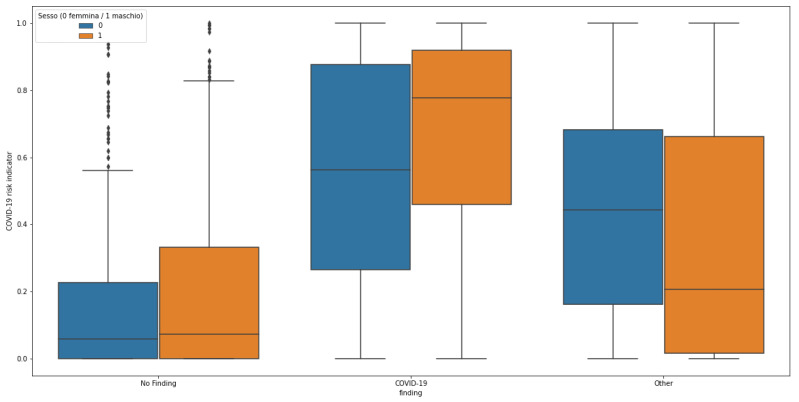
Distribution of the predicted COVID-19 risk indicator for cases belonging to the three classes, divided by gender. Blue boxes are referring to female, while orange ones are to male.

**Figure 10 diagnostics-12-00570-f010:**
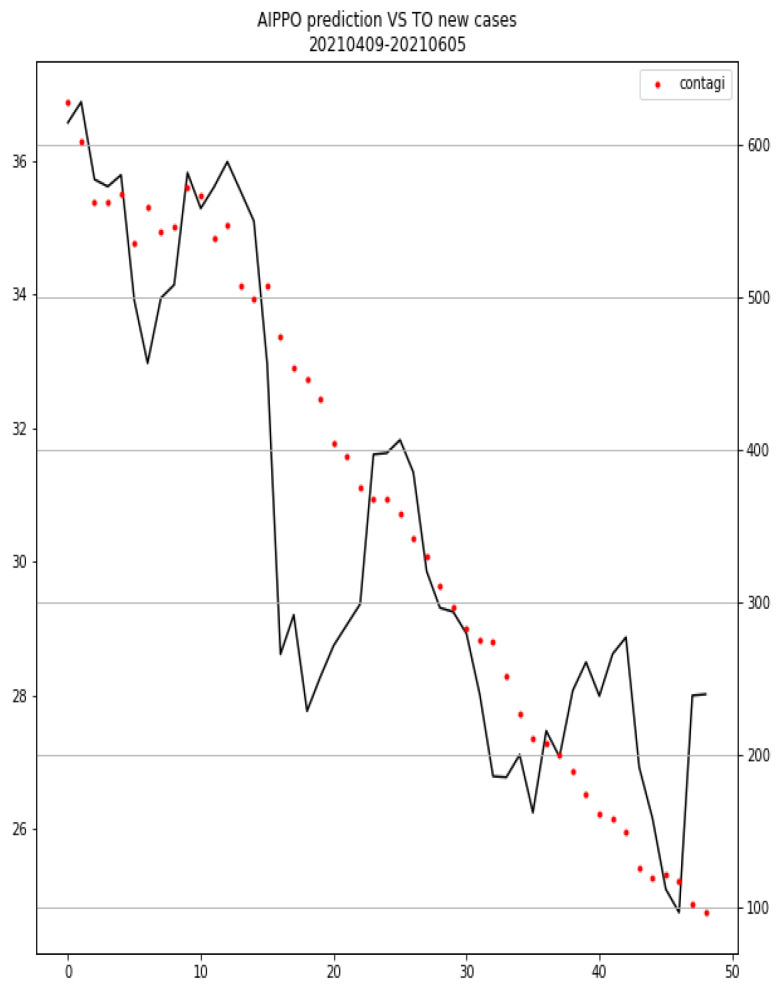
The 7 day rolling average comparison between AIppo COVID-19 score and Provincia di Torino daily new cases. Solid line indicates AIppo prediction, dots local new cases. Primary vertical axis (left): 7 day rolling average for AIppo score (in percentage), secondary vertical axis (right): 7 day rolling average for local new cases, horizontal axis: days.

**Table 1 diagnostics-12-00570-t001:** Metrics estimated for performance analysis.

Metric	Description	Interpretation
Mean Average Precision—MAP	Given the sorting induced by the COVID-19 indicator, this metric indicates, on average, the portion of positive cases placed before another positive one	Higher values indicate the positive cases are placed on top of the diagnostic queue
Sensitivity	Portion of COVID-19 positive individuals correctly identified by the system (COVID-19 risk indicator higher than a threshold)	Higher values indicate the method can identify a larger majority of positive cases
Specificity	Portion of COVID-19 negative individuals correctly identified by the system (COVID-19 risk indicator lower than a threshold)	Higher values indicate the method can identify a larger majority of negative cases

**Table 2 diagnostics-12-00570-t002:** Age statistics of the positive and negative groups. Except for *p*-values, all figures are in years.

Age Group	COVID-19 Prevalence Female	COVID-19 Prevalence Male	Chi-Square Test *p*-Value
<50 years	47.50%	31.71%	0.0335
50–59 years	45.45%	69.09%	0.0002
60–69 years	19.44%	42.62%	0.0001
70–79 years	44.07%	40.00%	0.0041
80–89 years	42.55%	43.75%	0.8273
≥90 years	68.75%	40.00%	0.0148

**Table 3 diagnostics-12-00570-t003:** Gender statistics of the positive and negative groups.

Gender	Total	COVID-19	Other	No Finding	Median Age	Age IQR
Female	231	41.64%	27.71%	30.65%	71	56–82
Male	305	44.67%	31.81%	23.52%	68	56–78

**Table 4 diagnostics-12-00570-t004:** Performance measured for AIppo system in phase 1.

Metric	Reference	AIPPO
MAP	43.79% (40–48%)	71.75% (63–81%)
Sensitivity	61.1% (55–67%)	78.23% (65–92%)
Specificity	63% (40–89%)	64.20% (63–66%)

**Table 5 diagnostics-12-00570-t005:** MAP performance calculated for each age group.

Age Group	N	COVID-19 Positive Prevalence	Median MAP
<50 years	81	39.51%	0.8304
50–59 years	88	60.23%	0.9289
60–69 years	97	34.02%	0.8546
70–79 years	144	41.67%	0.6704
80–89 years	95	43.16%	0.6784
≥90 years	31	54.84%	0.7331

**Table 6 diagnostics-12-00570-t006:** MAP performance calculated for each gender.

Gender	N	COVID-19 Positive Prevalence	Median MAP
Female	231	42.42%	0.6679
Male	305	45.25%	0.7481

**Table 7 diagnostics-12-00570-t007:** COVID-19 risk indicator grouped by age group and gender for COVID-19 positive patients. In bold, significant *p*-values.

Age Group	Females	Median Prediction	Prediction IQR	Males	Median Prediction	Prediction IQR	*p*-Value
<50 years	19	0.4670	(0.3929, 0.726)	13	0.7446	(0.3988, 0.9081)	0.00172
50–59 years	15	0.5557	(0.1367, 0.8058)	38	0.7903	(0.4072, 0.9401)	0.00000
60–69 years	7	0.8181	(0.5145, 0.9475)	26	0.8249	(0.5152, 0.917)	0.29681
70–79 years	26	0.6990	(0.1836, 0.9179)	34	0.7873	(0.5518, 0.9157)	0.04494
80–89 years	20	0.5214	(0.2658, 0.879)	21	0.6652	(0.4248, 0.875)	0.03575
≥90 years	11	0.5654	(0.3055, 0.7996)	6	0.5450	(0.4761, 0.6407)	0.28172

## Data Availability

The data used for the evaluation presented in this study are not publicly available due to privacy issues.

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
