# Peer review of "Convolutional Neural Network-Based Automatic Analysis of Chest Radiographs for the Detection of COVID-19 Pneumonia: A Prioritizing Tool in the Emergency Department, Phase I Study and Preliminary “Real Life” Results"

_diagnostics, 2022, doi:10.3390/diagnostics12030570_

Round 1

Reviewer 1 Report

This manuscript is well written on a trending topic. However, this reviewer recommends amending some remarks in the method section and discussion. The amendments will increase the quality of the manuscript.

Methods section

This reviewer would like the authors to include the variable severity "score" in the patients classified as COVID-19 positive; the inclusion of this variable would be extremely valuable for the readers. Also, fundamentals are to mention the method they use to classify the severity. 

A second matrix that can measure the performance besides MAP could be the correlation value between the calculated severity and the age of patients and compare these correlations values between the CNN and the senior radiologists.

There have been differences in correlation between age and severity of Italian and Mexican populations following the same methodology. Please include the reference:

Correlation between Chest X-Ray Severity in COVID-19 and Age in Mexican-Mestizo Patients: An Observational Cross-Sectional Study. Biomed Res Int. 2021 Apr 29;2021:5571144. doi: 10.1155/2021/5571144. PMID: 33997012; PMCID: PMC8090453.

Comment these findings in the discussion of you paper, because, then the CNN should be questioned if its classification methods calculated the same MAP in separed groups: male, females and age groups.

Authors have sufficient sample size, can they create some groups and compare groups to discard any differences in the classification power of the CNN?

Considering to supplement or substitute Tables 2 and 3 with graphs or a more informative table with significance comparison (see examples of figures 2 and 3 from the recommended reference).

Reviewer 2 Report

The authors proposed a CNN based tool for automatic detection of COVID-19 pneumonia. The results look promising, but I hope authors could provide some clarification regarding to my doubts.

  1. The introduction is not enough and vague. Authors should include more related research of using CNN for early detection of COVID-19.

  1. In Figure 1, DNN is responsible for extracting features from new images. Author mentioned the weights are transferred from the ImageNet task. So how authors fine-tune the model on CXR data? Please add more details here.

  1. In Figure 1, f represents the database to store previous extracted features. Are they the features extracted from ground truth of CXR images? Please explain.

  1. In the preprocessing stage, how was the U-Net lung segmentation model be used here? If a segmentation model is used, the input should be a binary image but from Figure 1, the input of DNN seems to be an original image with only being copped.

  1. How did authors finally decide which group the new CXR image is belong to? Is the similarity calculated based on group? Please add more details here.

  1. In Table 1, MAP is calculated for each class but why the interpretation is: “higher MAP indicate the positive cases are places on top of the diagnostic queue“? Please explain more here.

Reviewer 3 Report

Dear editor, dear authors

The manuscript Diagnostics-1549238 from the field of deep learning in medical image analysis tests a classifier for the discriminative ability of COVID-19 pneumonia suspect thorax x-ray images.

The tool is properly designed, based on a classical U-net, and the training/testing procedure is based on a large dataset with balanced numbers of positive and negative images. 

The major issue of this manuscript that hampers understanding and demotivates the reader is the insufficient use of English. Major grammar mistakes, as well as improper word selection, reduce the quality of an otherwise correct scientific work. Some examples: 

  • Abstract: "positive dataset" == ?? Do you mean data from SARS Cov2 positive patients? Please formulate your text properly.
  • "real life", not "real world" 

The reviewer demands a professional English medical language proof with the appropriate invoice to be submitted with the corrected manuscript version.

Best regards

Round 2

Reviewer 1 Report

The authors have appropriately addressed the remarks stated by this reviewer.

Reviewer 2 Report

The manuscript after revision has been improved a lot.